

# New Particle Formation and impact on CCN concentrations in the boundary layer and free troposphere at the high altitude station of Chacaltaya (5240 m a.s.l.), Bolivia

C. Rose[1], K. Sellegri[1], I. Moreno[2], F. Velarde[2], M. Ramonet[3], K. Weinhold[4], R. Krejci[5], M. Andrade[2], A. Wiedensohler[4], P. Ginot[6, 7] and P. Laj[6]

[1] Laboratoire de Météorologie Physique CNRS UMR 6016, Observatoire de Physique du Globe de Clermont-Ferrand, Université Blaise Pascal, 24 avenue des Landais, 63171 Aubière, France

[2] Universidad Mayor de San Andres, LFA-IIF-UMSA, Laboratory for Atmospheric Physics, Campus Universitario Cota Cota calle 27, Edificio FCPN piso 3, Casilla 4680, La Paz, Bolivia

[3] Laboratoire des Sciences du Climat et de l'Environnement, LSCE/IPSL, CEA-CNRS-UVSQ, Université Paris-Saclay, F-91191, Gif-sur-Yvette, France

[4] Leibniz Institute for Tropospheric Research, Permoserstr. 15, 04318 Leipzig, Germany

[5] Department Environmental Science and Analytical Chemistry (ACES), Atmospheric Science Unit, Stockholm University, S 10691 Stockholm, Sweden

[6] Laboratoire de Glaciologie et Géophysique de l'Environnement, UMR 5183, UGA/CNRS, Grenoble, France

[7] Observatoire des sciences de l'Univers de Grenoble, UMS 222, IRD/UGA/CNRS, Grenoble, France

## Abstract

Global models predict that new particle formation (NPF) is, in some environments, responsible for a substantial fraction of the total atmospheric particle number concentration and subsequently contribute significantly to cloud condensation nuclei (CCN) concentrations. NPF events were frequently observed at the highest atmospheric observatory in the world, Chacaltaya (5240 m a.s.l.), Bolivia. The present study focuses on the impact of NPF on CCN population. Neutral cluster and Air Ion Spectrometer and mobility particle size spectrometer measurements were simultaneously used to follow the growth of particles from cluster sizes down to ~2 nm up to CCN threshold sizes set to 50, 80 and 100 nm. Using measurements performed between January 1 and December 31 2012, we found that 61% of the 94 analysed events showed a clear particle growth and significant enhancement of the CCN-relevant particle number concentration. We evaluated the contribution of NPF events relative to the transport of pre-existing particles to the site. The averaged production of 50 nm particles



during those events was 5072 cm$^{-3}$, and 1481 cm$^{-3}$ for 100 nm particles, with a larger
contribution of NPF compared to transport, especially during the wet season. The data set was
further segregated into boundary layer (BL) and free troposphere (FT) conditions at the site.
The NPF frequency of occurrence was higher in the BL (48%) compared to the FT (39%).
Particle condensational growth was more frequently observed for events initiated in the FT,
but on average faster for those initiated in the BL, when the amount of condensable species
was most probably larger. As a result, the potential to form new CCN was higher for events
initiated in the BL (67% against 56% in the FT). In contrast, higher CCN number
concentration increases were found when the NPF process initially occurred in the FT, under
less polluted conditions. This work highlights the competition between particle growth and
the removal of freshly nucleated particles by coagulation processes. The results support model
predictions which suggest that NPF is an effective source of CCN in some environments, and
thus may influence regional climate through cloud related radiative processes.
**1   Introduction**
Atmospheric aerosol particles are known to affect air quality, health (Seaton et al., 1995) and
climate. Beside their direct interaction with the solar and telluric radiations, aerosol particles
also act as condensation nuclei for cloud droplets. Cloud effects such as cloud albedo
(Twomey, 1977) and lifetime (Albrecht, 1989) constitute the largest uncertainty in the
estimation of the radiative forcing of the Earth's atmosphere (IPCC, 2013).
The interaction between aerosol particles and the formation of warm clouds relies on the
ability of the particles to serve as cloud condensation nuclei (CCN), which depends on the
water vapour supersaturation, particle size distribution and also the chemical composition
(e.g.: Roberts et al., 2010; Wex et al., 2010; Asmi et al., 2012). Besides the processing of
primary particles, other CCN sources were identified, such as regional new particle formation
(NPF) events (Kerminen et al., 2012).
NPF is a frequent atmospheric phenomenon including the formation of nanometer-sized
clusters from gaseous precursors and their subsequent growth to larger sizes (eg. Kulmala and
Kerminen, 2008). Typical growth rates between 1.8 and 10.7 nm h$^{-1}$ were found for particles
in the range 1.5 – 20 nm (Yli-Juuti et al., 2011), meaning that a few hours to a few days are
needed for nucleated particles to grow to CCN sizes, around 50-150 nm. The chance for these
clusters to grow to CCN sizes strongly depends on the competition between condensational
growth and their removal by coagulation onto pre-existing particles.





During the last few years, several global model investigations were dedicated to the study of
the CCN-size aerosol production attributed to atmospheric NPF (Makkonen et al., 2012;
Merikanto et al., 2009; Reddington et al., 2011; Spracklen et al., 2008). While the outcomes
of these different models may vary according to the way they treat NPF and aerosol particle
processes (Lee et al., 2013), most of them show an enhancement of the CCN number
concentration due to NPF, both in the boundary layer (BL) and in the free troposphere (FT).
Based on the study by Makkonen et al. (2012), predictions of the present day annual global
average CCN concentration in the BL show almost a fivefold increase when taking into
account NPF. According to Merikanto et al. (2009), 45% of global low-level cloud CCN at
0.2% supersaturation originate from nucleation, and 35% have been formed in the free and
upper troposphere. Slightly contrasting results are provided by Reddington et al. (2011) using
the global model GLOMAP against measurements conducted at 15 European ground based
stations in the frame of the EUCAARI project. Reddington and co-workers found that CCN-
sized particle concentrations in the BL were mainly driven by processes other than NPF,
which contributed significantly to the CCN budget at little less than a quarter of observational
sites included in the study.
However, observations to validate these predictions are scarce, especially for the FT, where
measurements are often technically challenging. In this context, the purpose of the present
study is to estimate the contribution of NPF to CCN formation at the station of Chacaltaya
(5240 m a.s.l., Bolivia) with a special attention in differentiating the CCN number
concentrations attributed to NPF occurring at the station from those attributed to particle
transport to the site. This analysis was performed using an indirect method based on the NPF
event classification previously reported by Rose et al. (2015) and particle number size
distribution measurements in the range 10-500 nm. In addition to global CCN number
concentrations, a more detailed analysis of NPF and subsequent CCN production in the BL or
in the FT is also reported.
**2   Measurements and methods**
**2.1   Observation site and instruments**
Aerosol particle number size distributions, together with routine meteorological parameters,
were measured at the Chacaltaya GAW station, located in a range of the Bolivian Andes at the



summit of Mount Chacaltaya (16°21.014' S, 68°07.886' W), 30 km North of La Paz (2
million inhabitants).
The mobility distribution of charged particles and ions ($3.2 - 0.0013$ $cm^2V^{-1}s^{-1}$) and the size
distribution of total particles ($2 - 42$ nm) were measured by a Neutral cluster and Air Ion
Spectrometer (NAIS, Airel Ltd., Mirme and Mirme, 2013). The NAIS sampled the ambient
aerosol through an individual non-heated short inlet ($\sim 50$ cm) with a 5 minute time
resolution. Since the NAIS was likely to overestimate particle number concentrations above
20 nm (Manninen et al., 2016), particles in the range from 20 nm to CCN relevant sizes were
preferentially measured using a  mobility particle size spectrometer type TROPOS-SMPS
(Wiedensohler et al., 2012). The SMPS operated behind a Whole Air Inlet equipped with an
automatic dryer.
More details on the measurement site as well as the instrumental setup and the data quality
assurance can be found in  Rose et al. (2015) and Andrade et al. (2015).

## 2.2  Indirect method for the estimation of the NPF contribution to the CCN production

In absence of direct CCN measurements at Chacaltaya, the contribution of NPF to CCN
production was estimated from the continuous monitoring of the particle number size
distribution. This indirect method was first introduced by Lihavainen et al. (2003) and has
already been used in several other studies (Asmi et al., 2011; Kerminen et al., 2012; Laakso et
al., 2013; Laaksonen et al., 2005).
The basic hypothesis is that the lower cloud droplet activation diameter of aerosol particles is
in the range 50-150 nm for the usual supersaturations encountered in natural clouds (Asmi et
al., 2011, 2012; Komppula et al., 2005) including those forming at altitudes up to 3580 m
a.s.l., as observed at the Jungfraujoch station (Switzerland) (Hammer et al., 2014; Jurányi et
al., 2011). Although these conditions might be slightly different from those found in clouds
forming above 5000 m, we assume that on a first approach the CCN sizes previously
mentioned apply the same way at such altitudes. Thus, CCN number concentrations are
assimilated to a range of three different CN concentrations: hereafter, $CCN_{high}$ and $CCN_{low}$
refer to the higher and lower limits of the CCN concentration estimated from the number
concentrations of particles larger than 50 nm and 100 nm, respectively; as additional
information, an intermediate CCN concentration ($CCN_{med}$) was deduced from the number




concentration of particles larger than 80 nm. The CCN production during an event was
obtained from the comparison of the CCN concentration $N_{init}$ prior to and the maximum CCN
concentration $N_{max}$ during the event. For each particle diameter range, $N_{init}$ is defined as the
30 minute average concentration obtained at $t_{init}$, when nucleated particles reach the threshold
size, whereas $N_{max}$ is the 30 minute average concentration calculated when the CCN
concentration reaches a maximum during an event, at $t_{max}$. The determination of $N_{init}$ and $N_{max}$
is depicted on Fig. 1.
In addition to this first analysis classically used in the literature (that will be used in Section
3.1.1), further calculations are needed to take into consideration the geographical specificity
of the site. Indeed, if NPF contributes to the formation of potential new CCN, particles
transported to the site by diurnal forced or heat convection might also, in parallel, lead to an
apparent increase of the CCN number concentration. Thus, the CCN number concentrations
estimated using the methodology previously described, and attributed to NPF in a first
approach, might in fact result from both NPF and transport. The transport of particles to the
site is taken into account in Section 3.1.2., based on the hypothesis that similar number
concentrations of particles are transported to the site on event and non-event days. The
contribution of NPF to the production of new CCN was thus estimated from the difference
between the median CCN increases obtained on event (contributions from NPF and transport)
and non-event days (transport only). The hypothesis that the specific environmental
conditions on which NPF occurs are not favouring the transport from lower atmospheric
layers is not necessarily true, as NPF events were favoured during clear sky conditions (Rose
et al. 2015). Thus there is likely a bias towards an underestimation of transport from lower
atmospheric layers due to the fact that cloudy days are over-represented for non-event days.
This likely leads to an overestimation of the contribution of NPF to CCN number
concentrations. Nevertheless, this correction was never applied in the past, and certainly helps
approaching a more realistic view of the real contribution of NPF to CCN number
concentrations.
It is worth noticing that this indirect method based on particle size only provides estimations
of potential CCN concentrations instead of real concentrations as measured by CCN chambers
(Roberts and Nenes, 2005). However, for simplicity, we refer to these potential CCN as CCN
hereafter.





The selection of the NPF events to be analyzed was performed based on the following
criteria: only those NPF events with a clear particle growth (i.e. type I event following the
classification by Hirsikko et al., 2007) were considered and the days showing an eventual
contribution from NPF events triggered the day before were rejected. Regarding this aspect,
our analysis is thus a lower limit of the contribution of NPF to CCN-size relevant aerosol
concentrations.
**2.3   Method to assess the influence of the boundary layer in Chacaltaya**
In order to assess whether the site is under the influence of the planetary boundary layer or the
low free troposphere we employed the hourly-averaged value of the standard deviation of the
horizontal wind direction ($\sigma_\theta$).
The value of $\sigma_\theta$ has been extensively used in air pollution monitoring (EPA, 2008; Mitchell,
1982; Mitchell and Timbre, 1979; Weber, 1997) and dispersion models as an indicator of the
stability of the lower atmosphere. Instable atmospheric conditions produce turbulence and
therefore high wind variability. Conversely, low wind variability due to stable conditions
produces low $\sigma_\theta$ values. In Chacaltaya, $\sigma_\theta$ was used from a mountain perspective, .i.e.
assuming that turbulent conditions ($\sigma_\theta \geq 12.5$) reflect the influence of the BL at the observatory
and, contrarily, that non-turbulent (or stable) conditions are equivalent of being in the FT ($\sigma_\theta$
<12.5).
In Chacaltaya, $\sigma_\theta$ is obtained at the summit (5380 m a.s.l., 10 m above the surface) by means
of a wind vane and propeller (Young 05103) and processed directly on a CS-CR1000
datalogger. Sigma theta is defined as the standard deviation of the horizontal wind direction
itself according to Eq. (1), but its value is approximated by the Yamartino (1984) single-pass
method (set of Eq. (2)) directly in the datalogger.
$$\sigma_\theta = \left[ \frac{\sum_{i=1}^{N} (\theta_i - \theta_A)^2}{N-1} \right]^{\frac{1}{2}}$$   (1)
where $\theta_i$ is the instantaneous wind direction and $\theta_A$ the average wind direction.



$$\sigma_\theta = \arcsin(\varepsilon)\left[1 + \left(\frac{2}{\sqrt{3}} - 1\right)\varepsilon^3\right]$$

$$\varepsilon \equiv \sqrt{1 - (S^2 + C^2)}$$

$$S = \frac{1}{N}\sum_{i=1}^{N}\sin\theta_i \qquad\qquad (2)$$

$$C = \frac{1}{N}\sum_{i=1}^{N}\cos\theta_i$$

The synoptically driven change of wind direction may affect the calculation of $\sigma_\theta$ for short
time periods. This low-frequency horizontal wind oscillation is called "meandering" and may
produce overestimation of $\sigma_\theta$ during situations of low wind speed ($\leq 2\text{m.s}^{-1}$), which usually
take place during daytime in Chacaltya. Therefore, 15-min averaged values are calculated
offline according to Eq. (3) to avoid wind meandering effects.
$$\sigma_{\theta(1-hr)}{}^2 = \frac{\sigma_{\theta(15)}^2 + \sigma_{\theta(30)}^2 + \sigma_{\theta(45)}^2 + \sigma_{\theta(60)}^2}{4} \qquad\qquad (3)$$
where every $\sigma_{\theta(15x)}$ equation is a 15-minute deviation of the wind direction.
The threshold set for stable FT conditions is $\sigma_\theta \geq 12.5$, following Mitchell's recommendations
(1982). In Chacaltaya, FT conditions take place usually during night-time and before sunrise,
as it would be expected for mountain sites. Nevertheless, in many cases $\sigma_\theta$ values lower than
18 are observed in a persistent pattern (more than 4 hours of this condition). This may
indicate the existence of a residual or interface layer (IL). This intermediate layer would not
correspond neither to the FT nor the proper BL. Moreover, during the wet season, convective
and unstable conditions produce more turbulence at the site, shifting the $\sigma_\theta$ towards higher
values, typically below 18. Therefore other secondary site specific thresholds are applied,
namely 18 and 22.5.
Obtained hourly dataset is then checked for consistency, in particular with black carbon
measurements, and the following smoothing is applied. We establish a 4-hour window (2h
before and 2h after the data point of interest) into which the following criteria are applied:
• If the $\sigma_\theta$ value is lower than 12.5 (classified as FT), but if it is the only data point in
the 4-hour window, it is not considered as FT and it is reclassified as an IL point
instead.
• If the $\sigma_\theta$ value is lower than 18 and 75% of the points in the 4-hour window are lower
than 12.5, the point is classified as a FT point (stable).





- If the $\sigma_\theta$ value is lower than 22.5 and 75% of the points in the 4-hour window are lower than 18, the point is classified as an IL point (this takes place mostly during the wet season).

## 3 Results

### 3.1 CCN formation from NPF

### 3.1.1 Absolute and relative CCN production during NPF events

During the measurement period January 1 to December 31 2012, 147 days showing type I NPF events were detected: 112 during the dry season, from May to October, and 35 during the wet season, from November to April (Rose et al., 2015). Because of missing data of particle number size distribution measurements, only 94 of them were further analysed (75 from the dry season and 19 from the wet season).

Over the whole year, 61% of the studied NPF events were observed to grow to CCN-relevant sizes, and it is worth noticing that at Chacaltaya, when particles reached the lowest activation diameter, i.e. 50 nm, they systematically grew up to at least 100 nm. During the wet season, the frequency of aerosol particles originating from NPF event and reaching CCN sizes was higher compared to the dry season (79 % and 56%, respectively). This last observation can be ascribed to the larger growth rates which were detected during the wet season, being on average enhanced by a factor 1.7 compared to the dry season (Rose et al., 2015).

The results reported by Asmi et al. (2011) for Pallas (560 m a.s.l., Finland) using similar methodology slightly contrast with these observations. Indeed, the ability of NPF particles to contribute to the CCN number concentration showed a seasonal variation but also decreased with increasing activation diameter. This might be explained by a decreasing availability of condensing vapours over the course of the particle growth time period. At Chacaltaya, the availability of condensing gases appears to increase over a large time period, sometimes reaching concentrations that trigger a second (and third) nucleation event during the same day, in spite of the raising condensable sink due to the first nucleation event (Rose et al. 2015). Coagulation processes however lead to a decrease of $CCN_{low}$ compared to $CCN_{high}$. This is illustrated on Figure 2.a, which shows, for the three threshold sizes and for each season, the median CCN concentration increase observed during NPF events and calculated as the difference between $N_{max}$ and $N_{init}$. Considering all type I event days over the whole



year, the median number concentration of new CCN produced by NPF during an event was
5072 cm$^{-3}$ for CCN$_{high}$, 2254 and 1481 cm$^{-3}$ for CCN$_{med}$ and CCN$_{low}$, respectively. The
number concentration of new CCN was on average higher during the dry season, especially
for CCN$_{high}$.
Corresponding relative increases in CCN number concentration were calculated as the ratio of
the absolute increases previously reported over N$_{init}$, i.e. the 30 min average CCN number
concentration measured when nucleated particles initially reach the threshold sizes (Fig. 2.b).
NPF events were found to increase CCN concentrations by 168 to 996% at Chacaltaya, with
no clear differences between seasons or threshold sizes.
One should note that when several consecutive type I events were detected on a same day
(this occurred on 7 occasions), it was complex to extract the contribution of each individual
event, so the calculated CCN production was the result of the contribution of all events as a
whole. During multiple events days, the median number concentration of CCN produced was
on average 1.7 times higher compared to single type I event days.
As previously mentioned, similar methodology was used in previous studies to evaluate the
contribution of NPF to the CCN concentration. The average absolute CCN production from
NPF events at Chacaltaya is lower compared to that reported by Laaksonen et al., (2005) at
the station of San Pietro Capofiume located in the polluted region of the Pô valley (11 m
a.s.l., Italy): on the basis of 304 NPF events, the average number of new CCN produced
during an event are 7.3×10$^3$ cm$^{-3}$ and 2.4×10$^3$ cm$^{-3}$, for CCN$_{high}$ and CCN$_{low}$, respectively. In
contrast, the values from both Chacaltaya and San Pietro Capofiume are significantly higher
than those reported by Kerminen et al. (2012) for the stations of Botsalano (1420 m a.s.l.,
South Africa), Vavihill (172 m a.s.l., Sweden), Pallas and Hyytiälä (182 m a.s.l., Finland).
Among these four sites, the highest CCN concentration increases are on average observed at
Botsalano (2500 cm$^{-3}$, 1400 cm$^{-3}$and 800 cm$^{-3}$ for CCN$_{high}$, CCN$_{med}$ and CCN$_{low}$,
respectively), whereas Pallas displays the lowest CCN production (1000 cm$^{-3}$, 250 cm$^{-3}$and
150 cm$^{-3}$ for CCN$_{high}$, CCN$_{med}$ and CCN$_{low}$, respectively). Corresponding relative increases in
CCN concentrations found in the literature are always larger than 100% but never exceed
400%, being thus on average significantly lower than those observed at Chacaltaya. However,
it is worth noticing that these contrasting results may arise from the various conditions that
are found at the different stations, especially regarding altitude and pollution levels, thus
influencing NPF both in terms of strength, spatial extend and temporal evolution.



The potential of NPF to contribute to CCN production at high altitude was more particularly
investigated by Pierce et al. (2012) at Mount Whistler (2182 m a.s.l., Canada), following a
different approach including calculations of the probability for freshly nucleated particles to
reach CCN relevant sizes. Based on a five event day period, they found that in absence of
high coagulation/condensation sinks, up to 24% of the newly formed clusters could grow to at
least 100 nm, thus forming potential CCN.
However, as previously mentioned, the vertical transport of aerosol particles from lower
atmospheric levels that takes place after the onset of sunrise concurrently to NPF may
represent a significant contribution to the increase of CCN-relevant size particle number
concentrations at these mountain sites. This aspect will be addressed in the next section, in
which the contribution of NPF is further compared with the CCN number concentration
increase resulting from the transport of particles to the site.
The seasonal and annual CCN productions due to NPF events were estimated from 1) the
average fraction of type I NPF events contributing to the formation of new CCN reported
above, 2) the frequency of occurrence of type I NPF events at the site and 3) the average CCN
number concentration increase measured for those type I events during which newly formed
particles reached the potential CCN activation diameter. As an example, the $CCN_{high}$
production during the wet season was calculated as follows:
$$CCN_{high-wet} = frac_{wet} \times tot\_nb_{wet} \times avg\_conc_{wet} = 79\% \times 35 \times 3070 = 8.48 \times 10^4 \, cm^{-3} \quad (4)$$
where, for each season, $frac$ is the fraction of NPF events producing CCN, $tot\_nb$ is the
total number of days showing type I events and $avg\_conc$ is the median number of new CCN
formed during an event. Similar calculations were done for each season and CCN class,
leading to the values reported in Table 1. The annual CCN production was calculated as the
sum of the seasonal productions.
Based on Table 1, the CCN production at Chacaltaya was higher during the dry season
compared to the wet season for all CCN classes, but especially for $CCN_{high}$, which was more
than 4 times higher compared to the wet season. The annual CCN production calculated at
San Pietro Capofiume is $3.4 \times 10^5$ cm$^{-3}$ and $1.1 \times 10^5$ cm$^{-3}$, for $CCN_{high}$ and $CCN_{low}$, respectively
(Laaksonen et al., 2005). These values are slightly lower than those obtained at Chacaltaya,
despite the fact that the median number of potential new CCN formed during an event is on
average higher in San Pietro Capofiume. This last observation can be ascribed to the high



NPF frequency at Chacaltaya, together with the significant fraction of type I events and high
growth rates (Rose et al., 2015).

### 3.1.2 Correction for the contribution of particles transported to the site

The aim of this section is to evaluate the contribution of particles transported to the site to the
total CCN concentration and give a revised estimation of the CCN production from NPF.
The number concentration of CCN transported to the site was calculated using the particle
number size distributions recorded on non-event days. These conditions were fulfilled on 108
days (23 and 85 during the dry and wet season respectively) but only 78 of them (22 from the
dry season and 56 from the wet season) were further analysed because of instrumental
failures. The median diurnal variation of $CCN_{high}$ obtained on non-event days and attributed to
transport is shown on Fig. 3, together with the median number concentrations obtained on
event days and ascribed to both NPF and transport (upper panel). Similar figures are reported
in the supplementary material for $CCN_{med}$ and $CCN_{low}$ (Figures S1 and S2). As previously
mentioned, the contribution of NPF to the production of new CCN was estimated from the
difference between the median $CCN_{high}$ increases obtained on event and non-event days and is
shown on Fig. 3 (lower panel).
Outside of the NPF hours, the CCN number concentrations are on average similar on event
and non-event days for all sizes and seasons. During the dry season, transport contributes to
$CCN_{med}$ and $CCN_{low}$ to the median level of 1139 and 863 cm$^{-3}$, which is similar to the
contribution of NPF (between 1229 and 784 cm$^{-3}$ depending on the season). In contrast,
$CCN_{high}$ attributed to NPF (3197 cm$^{-3}$) significantly exceeds the median number of particles
transported to the site (1610 cm$^{-3}$). During the wet season, NPF is likely to be the dominant
CCN source, with productions of 1950, 771 and 535 cm$^{-3}$ for $CCN_{high}$, $CCN_{med}$ and $CCN_{low}$,
respectively, compared to median concentrations attributed to transport which do not exceed
690, 404 and 321 cm$^{-3}$.
As expected, these revised NPF contributions are decreased compared to the values reported
on Fig. 2.a. As a consequence, the corrected seasonal and annual CCN productions ascribed to
NPF and reported in Table 2 are also decreased compared to the values shown in Table 1.
However, the revised NPF contributions still remain comparable or even higher than those
previously mentioned for other stations, and which probably also include CCN sources other
than NPF.



## 3.2 How layering influences growth to CCN-sizes

### 3.2.1 Occurrence of NPF in the different tropospheric layers

The purpose of this section is to further investigate NPF in terms of occurrence, event type and characteristics (particle formation and growth rate) regarding the location of the station in the tropospheric layers (i.e. BL, FT or IL) at the onset of the NPF process. The classification of air mass types into BL, IL and FT was obtained using the Pasquill-Gifford method, which uses the turbulence from the standard deviation of wind direction (Section 2.2).

389 NPF events previously discussed by Rose et al. (2015) were included in this analysis. For each event, the air mass type (BL, IL or FT) prevailing at the station was investigated on an hourly basis during the first steps of the NPF process, i.e. from the appearance of the newly formed clusters (< 3nm) to the time at which the concentration of 3-7 nm particles was maximum. Various scenarios were observed during this part of the NPF process, which on average lasted for 2.7±1.3 hours. These scenarios are listed, together with their frequency of occurrence, in Table 3. Scenarios S1- S3 refer to those days when the initial steps of the NPF process were observed to occur in the same atmospheric layer (respectively in the BL, in the IL and in the FT). In contrast, scenarios S4 – S9 depict the situations when BL dynamics lead to changing conditions in the course of the event, with a gradual evolution from BL to FT (S7 – S9) or vice versa (S4 - S6). S10 refers to events occurring in conditions changing randomly. Since multiple events were frequently detected at Chacaltaya, additional information regarding the occurrence of the scenarios as a function of the event position (first event, second event, third and following events) is also provided. For that purpose, single events and events occurring first on multiple event days were considered all together, while second and following events were considered in a second category. There was no information available regarding the classification into BL, IL and FT for 15 events.

Based on Table 3, constant conditions, i.e. scenarios S1 (BL conditions only), S2 (IL conditions only) and S3 (FT conditions only), were found in 57% of the observed single and first position events and 93% of the second and following events. In each case, scenario S1, corresponding to BL conditions, was the most frequent, representing 89% and 94% of the events initiated in constant conditions, respectively for single and first position events and for second and following ones. The fact that scenarios related to changing conditions were more frequently observed for single and first position events (41% compared to 7% for following




events), i.e. occurring earlier in the morning compared to following events, is mainly
explained by the development of the BL during the first part of the day, as shown on Fig. 4.
This figure also supports the fact that when changing conditions are observed, scenarios
starting in the FT, i.e. scenarios S4, S5, and S6, are the most probable for both seasons.
Among the multiple events days, 48 displayed consecutive events associated to scenarios
evolving in agreement with the BL dynamics shown on Fig. 4. The sequence S6-S1 was the
most frequent, observed on 23 days, following by the sequences S7-S1 (10 days) and S5-S1
(5 days). A unique sequence including three events was detected (S3-S6-S1).
NPF frequencies in the FT and in the BL were also deduced from the previous classification.
For that purpose, the analysis was focused on the time period 08:00 - 12:00 (Local), which
includes the most probable nucleation hours (Rose et al., 2015). 72 days (including both
event, non-event an undefined days) were rejected from the analysis because of missing
information regarding the location of the station in the tropospheric layers. Free tropospheric
conditions were detected during at least one hour on 122 days, and among these days, 48
showed NPF events initiated in the FT, leading to a NPF frequency of 39%. In contrast, the
station laid in the BL during at least one hour on 248 days, and among these days, 119
showed events starting in the BL, leading to a NPF frequency of 48%.
### 3.2.2 Event type and characteristics
An additional analysis concerning the event type (i.e. I, II or bump, Hirsikko et al., 2007) as a
function of the scenario was performed using the event classification from Rose et al. (2015).
The results of this analysis are shown on Fig. 5. More than half of the 77 events trigered in
the FT (scenarios S3, S4, S5 and S6, Table 3) were identified as type I events (39 events),
while types II and bump events were observed on 17 and 21 occasions, respectively, which
represent 22 and 27% of these scenarios. When considering the scenarios S3, S4, S5 and S6
independently from one another, we found that type I events were predominant when
changing conditions were detected (S4, S5 and S6), whereas they displayed similar
probabilities of occurrence as other event types in constant free tropospheric conditions (S3).
This observation suggests that the probability for type I events to occur is increased when
initial free tropospheric conditions are changing in the course of the events. This could be
explained by favorable conditions for the onset of nucleation events, followed by increased
input of condensable species from the BL promoting particle growth. However, this
hypothesis must be considered with caution regarding the limited number of events occurring





under scenario S3. Events starting at the interface between the BL and the FT are not
frequently observed (S2 and S7, 29 events), however, almost 50% of them (14 events) belong
to class I. Most of the remaining events occur under scenario S1, in the BL, with comparable
number of events belonging to class I and II (87 and 91 events, thus representing 40 and 42%
of scenario S1, respectively).
In order to further characterize the NPF events in the different atmospheric layers, statistics
regarding the formation rate of 2 nm particle and the growth rate (GR) in the size range 1-3
nm as a function of the scenarios were performed for type I events. As reported on Fig. 6,
NPF events initiated in the FT or at the interface between the BL and the FT show similar
particle formation and growth rates. Increased values are on average reported in the BL, with
higher variability, especially for the GR. Additional analysis was performed to investigate the
correlation between the GR in the size range 3-7 nm and the location of the station at the end
of the scenarios. However, because of an insufficient number of values for events occurring
under scenarios ending in the FT (scenarios S3 and S9, 4 values), these results will not be
further discussed.
We have shown so far that while higher NPF frequencies where found in the BL compared to
the FT, higher probabilities for type I events to occur were associated to scenarios starting in
the FT and ending in the BL or IL. However, when events belonging to class I are initiated in
the BL, they show on average higher particle formation and growth rates compared to those
started in the FT. Thus it is likely that on the one hand, higher amounts of gaseous precursors
usually associated with the BL could favor nucleation events of higher intensity and explain
both higher NPF frequencies and enhanced particle formation and growth rates. On the other
hand, cleaner conditions found in the FT at the very beginning of the NPF process may reduce
the sink for the newly formed clusters and favor their growth to larger sizes. This observation
suggests that the amount of condensable species could directly influence the occurrence of the
NPF process and determine the particle growth rate while the occurrence of the growth
process itself could rather depend on the strength of the particle sink. Overall, the difference
of occurrence frequency, nucleation rates and GR between FT and BL are not very large, and
we show that nucleation is initiated in the FT with a rather high frequency.
The purpose of the next section is now to investigate the impact of these NPF events on the
CCN number concentration in each of the atmospheric layers.





### 3.2.3 CCN production during NPF events in the different tropospheric layers

Based on the results discussed in section 3.1.1, 57 NPF event days showing particle growth up to CCN activation diameter were detected at Chacaltaya. 13 of them were not further analyzed due to missing information regarding the location of the station in the tropospheric layers. The remaining 44 days were all single type I event days, among which 31 were initiated in the BL, 10 in the FT, 2 at the interface between the BL and the FT and 1 in random conditions. The frequency of NPF contribution to the production of new CCN in the BL and in the FT was calculated as the ratio of NPF events growing to the CCN sizes to the total number of type I events occurring in each atmospheric layer, i.e. 46 in the BL and 18 in the FT. The resulting frequency of CCN production from NPF was 67% in the BL, being slightly higher compared to the FT (56%).

The number concentration of CCN formed during an event was also analyzed as a function of the air mass type (BL, IL, or FT) prevailing at the station (Table 4). Using the three threshold sizes, median CCN productions were comparable for events initiated in the BL and in the FT. In contrast, the third quartiles of $CCN_{med}$ and $CCN_{low}$ were higher for the events initiated in the FT.

The fact that the contribution of NPF to the formation of new CCN was more frequently observed for events initiated in the BL might be explained by faster particle growths sustained by higher amounts of condensable material, thus increasing the chances for particles to reach CCN sizes. The tendency for $CCN_{med}$ and $CCN_{low}$ to reach higher values when the NPF process was started in the FT can be due to smaller initial concentrations prior to the NPF event, and thus weaker coagulation associated to less polluted conditions in the FT.

### 4 Conclusion

In this paper, the contribution of NPF to the production of potential new CCN was investigated at the highest station in the world, Chacaltaya (5240 m a.s.l., Bolivia), between January 1 and December 31 2012.

Using potential CCN activation diameters 50, 80 and 100 nm, we found that 61% of the type I NPF events included in the analysis produced new CCN, with higher probabilities during the wet season (79%) explained by faster particle growth. Because of coagulation on pre-existing particles, the number concentration of CCN formed was observed to decrease with increasing activation diameter, but the frequency of particles reaching the highest potential CCN





activation diameter (100nm) was not reduced compared to the lowest CCN size (50 nm).
When comparing the CCN production from NPF with the number concentration of CCN
transported to the site, we found that NPF was on average responsible for the largest
contribution to the CCN concentration, especially during the wet season.
When segregating into BL and FT air masses sampled at the site, we found slightly higher
NPF frequency in the BL (48%) but still an important frequency of occurrence in the FT
(nucleation frequency of 39%). This observation is, to our knowledge, the first of its kind.
Particle growth was more frequently observed for events initiated in the FT but was on
average faster for events started in the BL, most probably because of increased amounts of
condensable vapours. As a result, the chance for particles to grow up to potential CCN
activation diameters was higher when the NPF process occurred in the BL. In contrast, the
impact of NPF initiated in the FT on CCN number concentrations was higher than for NPF
initiated in the BL, most likely because of the decreased pollution levels and weaker
coagulation sink. The previous observations clearly highlight the competition that exists
between particle growth and their removal by coagulation processes on pre-existing particles,
and thus the complex balance between sources and sinks that is required to observe the
formation of new particles and their subsequent growth to climate relevant sizes. Such
conditions are often fulfilled at Chacaltaya, where NPF seems to often play a dominant role in
the formation of new CCN.
*Acknowledgments*
*This work was performed within the framework of  ACTRIS2 (Aerosols, Clouds and Trace*
*gases Research Infra Structure Network) and was also supported by SNO CLAP, IRD, LEFE-*
*CHAT, OSUG@2020 as well as STINT and FORMAS funding agengies.*

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





1    Table 1 Estimation of the median seasonal and annual CCN productions from NPF.

|  | $CCN_{high}$ (cm$^{-3}$) | $CCN_{med}$ (cm$^{-3}$) | $CCN_{low}$ (cm$^{-3}$) |
|---|---|---|---|
| Dry season | $3.96 \times 10^5$ | $1.60 \times 10^5$ | $9.40 \times 10^4$ |
| Wet season | $8.48 \times 10^4$ | $4.98 \times 10^4$ | $3.90 \times 10^4$ |
| Whole year | $4.81 \times 10^5$ | $2.10 \times 10^5$ | $1.33 \times 10^5$ |

4    Table 2 Estimation of the median seasonal and annual CCN productions from NPF corrected

5    for the contribution of particles transported to the site.

|  | $CCN_{high}$ (cm$^{-3}$) | $CCN_{med}$ (cm$^{-3}$) | $CCN_{low}$ (cm$^{-3}$) |
|---|---|---|---|
| Dry season | $2.00 \times 10^5$ | $7.71 \times 10^4$ | $4.92 \times 10^4$ |
| Wet season | $5.39 \times 10^4$ | $2.13 \times 10^4$ | $1.48 \times 10^4$ |
| Whole year | $2.54 \times 10^5$ | $9.84 \times 10^4$ | $6.40 \times 10^4$ |



Table 3 Description of the scenarios concerning the location of the station in the troposphere
(boundary layer (BL), interface layer (IL) and free troposphere (FT)) during the first steps of
the NPF process. The total number of occurrence is provided for each scenario in the second
column. Since multiple events are frequently observed at Chacaltaya, a more detailed
classification including the event position is specified in the last two columns.

| Scenario | Description | Total number of occurrence | Single and first position events | Second and following events |
|----------|-------------|----------------------------|----------------------------------|-----------------------------|
| S1 | BL only | 217 | 100 | 117 |
| S2 | IL only | 9 | 6 | 3 |
| S3 | FT only | 12 | 7 | 5 |
| S4 | FT first, IL following | 13 | 13 | 0 |
| S5 | FT first, IL and then BL | 11 | 11 | 0 |
| S6 | FT first, BL following | 41 | 37 | 4 |
| S7 | IL first, BL following | 20 | 18 | 2 |
| S8 | BL first, IL following | 2 | 1 | 1 |
| S9 | BL first, FT following | 1 | 1 | 0 |
| S10 | Random conditions | 7 | 5 | 2 |



4  Table 4. CCN production as a function of the location of the station (BL or FT) at the onset of

5  the NPF process.

| Threshold CCN size | CCN increase for events started in the BL (cm$^{-3}$) | | | CCN increase for events started in the FT (cm$^{-3}$) | | |
|---|---|---|---|---|---|---|
| | 25$^{th}$ perc. | Median | 75$^{th}$ perc. | 25$^{th}$ perc. | Median | 75$^{th}$ perc. |
| 50 nm | 2556 | 5072 | 10110 | 3070 | 5137 | 9378 |
| 80 nm | 1155 | 2416 | 3919 | 1483 | 2138 | 5173 |
| 100 nm | 820 | 1518 | 2338 | 960 | 1447 | 3568 |

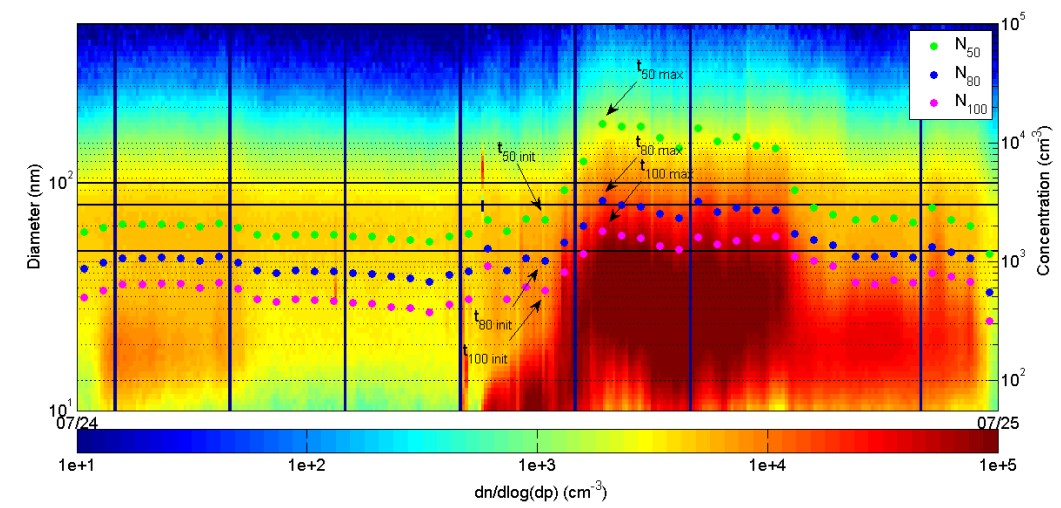

18  Fig. 1. Determination of the CCN concentration increase for the 3 threshold diameters (50, 80

19  and 100 nm). $t_{init}$ and $t_{max}$ denote, for each diameter, the times from which concentration

20  increases are calculated. July 24$^{th}$ 2012, Chacaltaya.





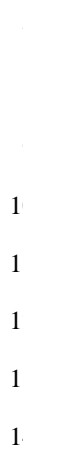

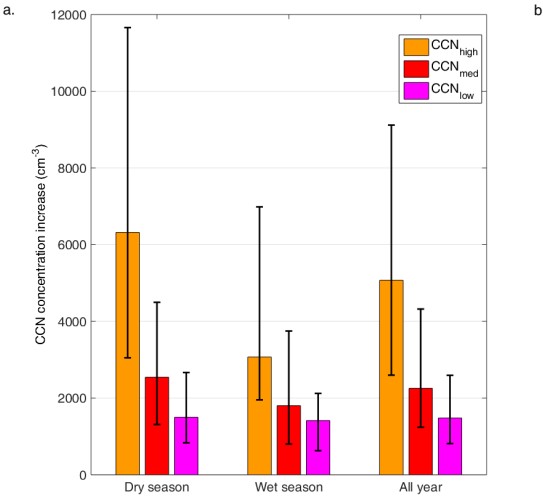

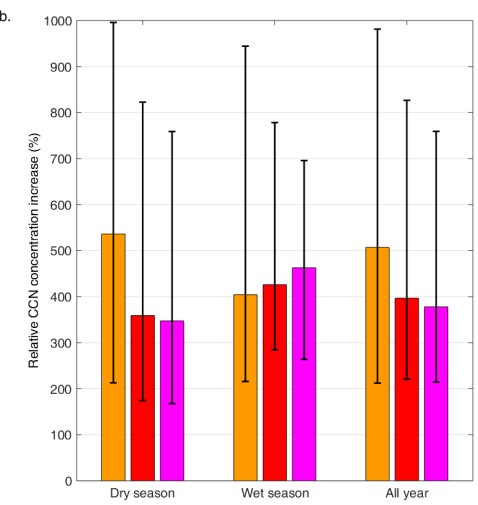

16   Fig. 2.  Median a. absolute and b. relative CCN productions observed during type I events for

17   the different activation diameters and seasons. Lower and upper limits of the error bars stand

18   for the 1$^{st}$ and 3$^{rd}$ quartile, respectively. Chacaltaya, 2012.





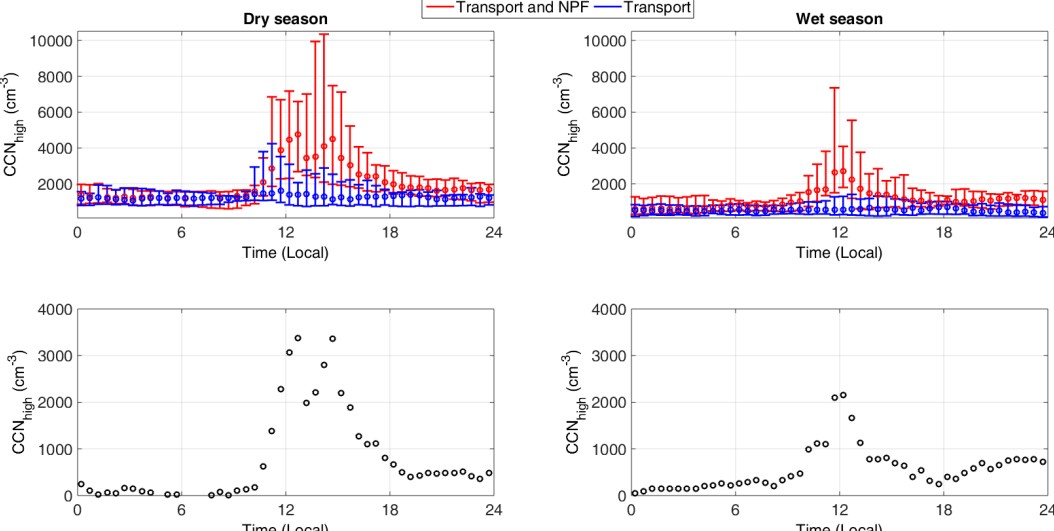

Fig. 3. Median diurnal variation of $CCN_{high}$ on event (upper panel, "Transport and NPF") and non-event days (upper panel, "Transport"). $CCN_{high}$ attributed to NPF (lower panel) is calculated as the difference of the concentrations recorded on event and non-event days. Lower and upper limits of the error bars stand for the 1st and 3rd quartile, respectively. Chacaltaya, 2012.





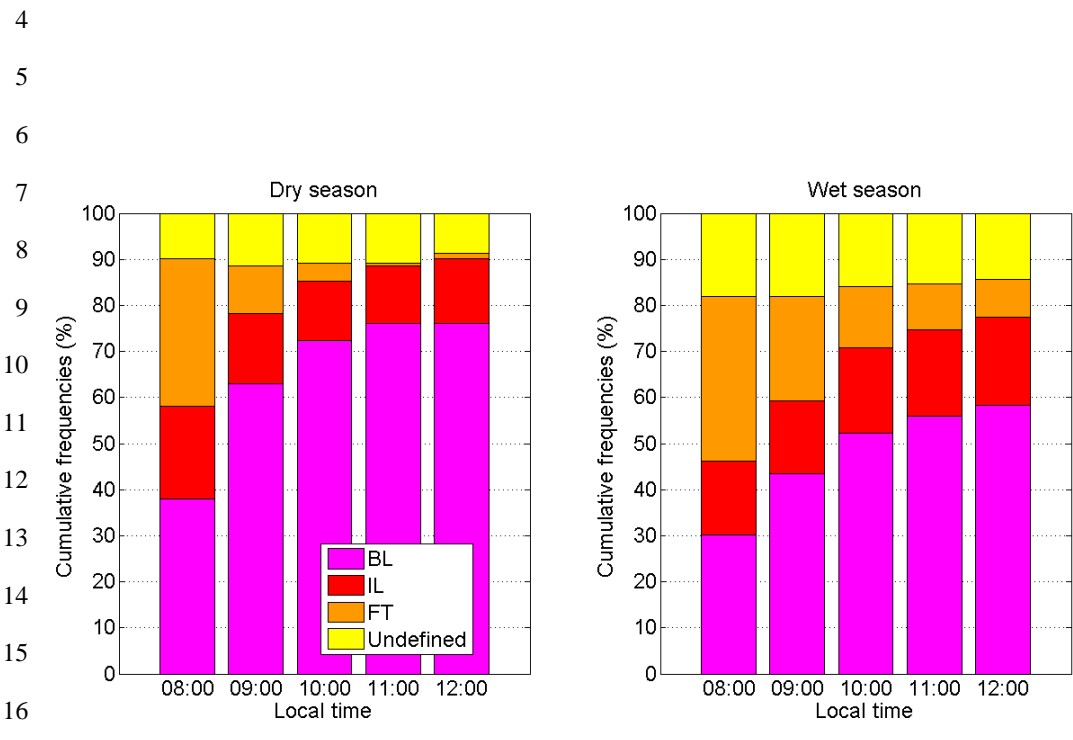

Fig. 4. Statistics on the location of the station in the tropospheric layers (boundary layer (BL),

interface layer (IL) and free troposphere (FT)) between 8:00 and 12:00 (Local), separately for

the dry and wet seasons. Chacaltaya, 2012.





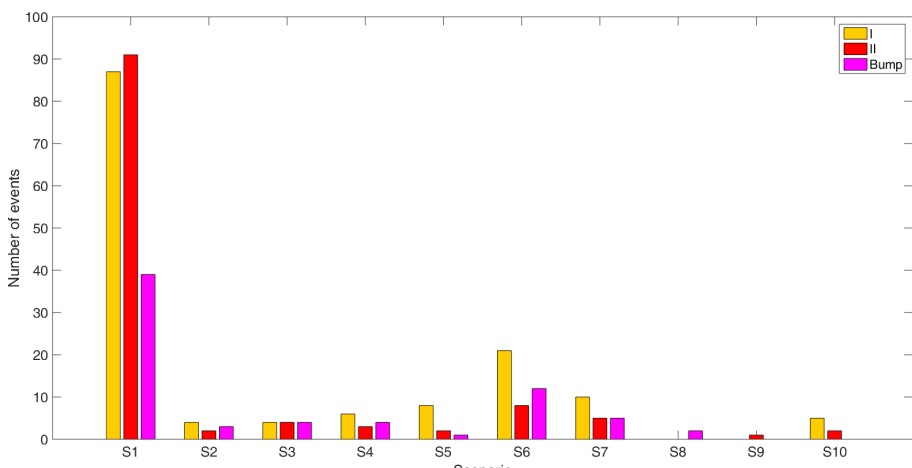

17  Fig. 5. Statistics on the event type (I, II or bump) as a function of the scenario describing the

18  location of the station in the tropospheric layers. Chacaltaya, 2012.





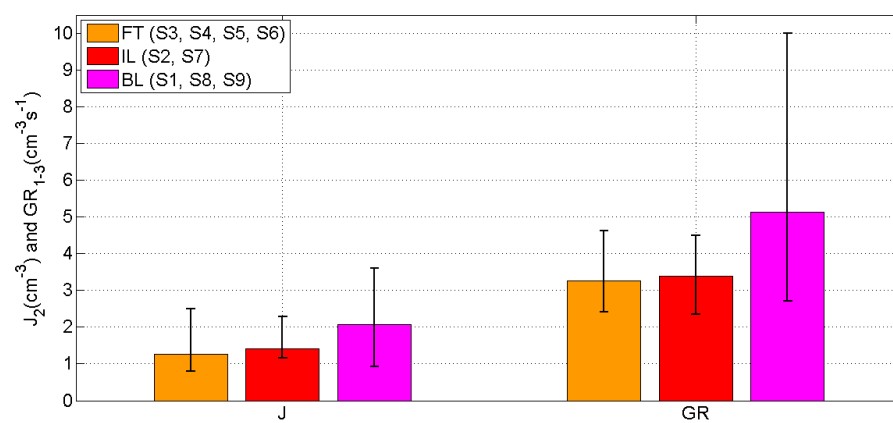

Fig. 6. Median particle formation and growth rates reported separately for type I events initiated in the FT (scenarios S3, S4, S5 and S6), at the interface between the BL and the FT (scenarios S2 and S7) and in the BL (scenarios S1, S8 and S9). Lower and upper limits of the error bars stand for the 1st and 3rd quartile, respectively. Chacaltaya, 2012.