# Peer review of "in the boundary layer and free troposphere at the high"

_Atmospheric Chemistry and Physics, 2016_

## Referee Comment (RC1) · Anonymous Referee #1 · 20 Sep 2016

Atmospheric Chemistry and Physics MS No.: acp-2016-696

Title: New Particle Formation and impact on CCN concentrations in the boundary layer and free troposphere at the high altitude station of Chacaltaya (5240 m a.s.l.), Bolivia

First author: C. Rose

The manuscript describes new particle formation (NPF) events observed at the Chacaltaya station in Bolivia and their contribution to the cloud condensation nuclei (CCN) population. The results are based on measurement done with the SMPS and NAIS where the CCN are particles with different sizes (50, 80 and 100 nm). The manuscript

also highlights the importance of studying this mechanism in order to better understand the influence of NPF in the CCN population. I consider that the data set is quite valuable (long term and remote location), however, the authors don't take full advantage of such data set. I therefore recommend the publication of this manuscript only after major revision.

I actually have three major concern listed here:

1- The CCN increase during NPF is confused with CCN increase due to NPF. The authors tried to differentiate these two points correcting for the transported CCN, but this is not enough, especcially because they don't correct for the growth of pre-existing particles. Since this is the major point of the paper I believe should be more solid. 2- The discrimination between FT and PBL is weak. The authors don't consider the history of these air masses which previous research has shown to be of paramount importance. 3- The section 3.2 is vague and beside some small details I don't see the take home message of this section.

Introduction: The introduction is well written; however, I think that some major studies have been forgotten or omitted. Recently, studies conducted at the Jungfraujoch appeared in Science and JGR-A. These studies should be mentioned as a comparison, especially in terms of CCN production, would be extremely valuable (Herrmann et al. 2015, Bianchi et al. 2016, Tröstl et al. 2016)

Page 2 Line 29: The reference Yli-Juuti is only about one site (Hyytiälä). I would recommend to consider the Manninen et al. (2010) EUCAARI paper which reports findings from 12 European sites.

Page 2 Line 30: Reference is needed.

Page 3 Line 17: "….However, observations to validate these predictions are scarce, especially for the FT…."

It is true that little information is available in this specific field of research. However, a

new study by Tröstl et al., (2016) has just been published in the Journal of Geophysical Research (Atmospheres) which (among other things) investigates the contribution of new particle formation to the CCN concentration in the Alps in some detail. I think a comparison to this work (Alps vs Andes) would be quite interesting, especially considering the general scarcity of similar research.

Page 4 Line 1,2: The authors underline the vicinity of the site to a city like La Paz that has a large population and is assumingly rather polluted. This fact seems weirdly underused in the study. Why not determine La Paz air masses to find out what (if any) effect polluted air masses have on NPF and CCN production? A backtrajectory analysis might actually be quite interesting.

Section 2.2 Indirect method for the estimation of the NPF contribution to the CCN production I'm not sure that this method fulfilled what the authors claimed. I agree with the fact that the CCN increased during NPF is not just a pure coincidence since the NPF precursors certainly also facilitate growth. However, it's not possible to distinguish the CCN formed by the NPF events with the growth of pre-existing particles during the same time. An easy way to fix that somehow would be to assume that ALL pre-existing particles to become CCN before any new particles. I.e. the number of particles below 100 nm before the event must be subtracted from the CCN100 you are now using and so far for the other sizes. This still would not account for, say, 90 nm particles that are transported to the site during NPF and grow above the threshold and contribute to CCNmax but it would be better than the current approach.

Page 4 Line 28: I found CCNhigh and CCNlow quite confusing. I would rather prefer to use the size of the particles, therefore, I would call them CCN50 and CCN100.

Page 5 Line 1: "...The CCN production during an event was obtained from the comparison of the CCN concentration Ninit prior to and the maximum CCN concentration Nmax during the event..."

I agree with the authors that this is the CCN production during an event. However,

the authors treat this as the CCN increase due to NPF. As already mentioned, CCN increase DURING and BECAUSE OF NPF are not the same thing. This distinction demands for clarity of concept and language whenever the topic is discussed. The manuscript in its current form, however, conflates those things. Besides the need for exact language, I actually think the issue can be addressed (to an extent) as lined out above.

Page 5 Line4 ". . ..tinit, when nucleated particles reach the threshold size. . .."

I don't think is possible to know that nucleated particles reached CCN size instead of larger particles that simply have grown above the threshold. The respective figure 1 actually shows that t_init isn't found as the text claims: if the text was true, then t_init_100 should be well after t_init_50 because growth takes time. In the figure, however, all t_init are the same. That means t_init is really just the time when CCN numbers start to increase. But that increase doesn't likely come from NPF. Figure 1 illustrates a further problem with this claim: t_init_100 is roughly 1.5...2 h after nucleation onset. If those were really newly nucleated particles, we would need growth rates of 50 nm/h. I find that hard to believe as such numbers have never (to my knowledge) been reported in the literature for atmospheric nucleation.

Page 5 Line 14: The authors acknowledge only partially the previous point. They say that the particles can be transported during that period but they still don't mention that small particles transported there can then grow to the threshold and being considered as formed by NPF. They also correct the transported particles by comparing NPF days with non-NPF days. This assumption is valid only if the physic dynamic is the same. If the NPF is triggered by the wind convection might be that during nucleation (more wind) the particles transported up there are more. This point needs further investigation or at least being commented.

Section 2.3 Method to assess the influence of the boundary layer in Chacaltaya.

To my understanding, this method only takes into account the local PBL influence at

the time of nucleation. It does little to actually describe the air mass in which nucleation occurs. Bianchi et al. (2016) have shown that strong PBL contact 1-2 days before NPF is crucial in the case of the Alps (Jungfraujoch). While conditions are certainly different in the Andes, there is no reason to believe that local wind conditions could accurately describe an air mass and its history which is what one must do to get a handle on PBL influence. There is a good body of literature dealing with the assessment of PBL influence. Much of it has been summarized in recent papers by Bukowiecki et al. (2016) and Herrmann et al. (2015).

Page 8 Line 14: "...when particles reached the lowest activation diameter, i.e. 50 nm, they systematically grew up to at least 100 nm..."

This statement is stronger than what the data seem to support. We don't know for certain that those are not pre-existing particles that simply did a bit of growing above the considered threshold, or do we?

Page 8 Line 16: "...aerosol particles originating from NPF event and reaching CCN sizes..."

Yet again the same problem that I think it should be fixed. I haven't seen any evidence that all those new CCN come directly from NPF, and, indeed, I find it highly unlikely: as long as there are pre-existing particles their chances to add to the CCN concentration are MUCH higher than the chances of newly formed particles.

Page 9 from Line 15 to Line 32: The paragraph lacking a message.

First the authors give us comparisons to sites that are hardly comparable to a 5000 m peak, and then they tell us in the last few lines that those comparisons are more or less pointless and I actually i would agree because these sites are just different and the comparison does not provide useful information. A comparison to Tröstl et al., (2016) might be more interesting, especially since those results are quite different.

Page 10 Line 8:

Sunrise is typically a well-defined point in time and not a process that has an onset.

Section 3.1.2 Correction for the contribution of particles transported to the site

I'm a bit concerned regarding this method to correct the contribution of particles transported. As mentioned earlier, this method is valid only in case every day we have the same physics and nucleation only depends on the vapors present. However, if nucleation is triggered by the wind coming up the valley than during nucleation we would have more transportation of big particles and therefore the correction method is not ideal. Would be nice to know what are the differences (Wind direction, wind speed etc..) during nucleation and during no nucleation where these background values is taken in account.

Section 3.2 How layering influences growth to CCN-sizes I do understand the need of knowing where the nucleation events take place and especially if this lead to a big production of particles in the free troposphere. However, I believe that dividing in 10 scenarios is a bit over exaggerated and probably not quite realistic. I think it would be better if the authors can simplify this section. I don't think that selecting more than 3 scenarios is feasible. In addition to that the split into different scenarios seems ill-conceived since most scenarios are quite irrelevant with very little occurrences. This might all be a nice exercise in data analysis but the text fails to tell us what the actual results are. What do we learn in this section apart from some minuscule details? This section has the feel of filler material and needs to be improved with a fair amount of actual substance.

Page 12 Line 4: ". . .regarding the location of the station in the tropospheric layers. . ."

I wonder if this is actually relevant at all? The NPF events are mainly driven by the air mass history and not so much by the atmospheric layer when the event begins.

Page 12 Line 8: "389 NPF events"

Is that a different data set?

Minor edits:

Figures: In general no need to state Chacaltaya at the end of every captions.

Figure 1: Why Tinit is not before the nucleation but already a after the start of the event? Please also describe the figure, Particle size distribution measured by…... and so on.

Refererences:

Bianchi, F., et al., (2016), New particle formation in the free troposphere: A question of chemistry and timing. Science, 10.1126/science.aad5456

Bukowiecki, N., et al., (2016), A Review of More than 20 Years of Aerosol Observation at the High Altitude Research Station Jungfraujoch, Switzerland (3580 m asl). Aerosol. Air Qual. Research, 10.4209/aaqr.2015.05.0305

Herrman, E., et al., (2015), Analysis of long-term aerosol size distribution data from Jungfraujoch with emphasis on free tropospheric conditions, cloud influence, and air mass transport. J. Geophys. Res., 10.1002/2015JD023660

Manninen, H.E., et al., (2010), EUCAARI ion spectrometer measurements at 12 European sites – analysis of new particle formation events. Atmos. Chem. Phys. 10.5194/acp-10-7907-2010

Tröstl, J., et al., (2016), Contribution of new particle formation to the total aerosol concentration at the high altitude site Jungfraujoch (3'580 m a.s.l., Switzerland). J. Geophys. Res., 10.1002/2015JD024637

―――――――――――――――――――

---

## Referee Comment (RC2) · Anonymous Referee #2 · 3 Nov 2016

GENERAL COMMENT

The manuscript presents an analysis of 12 months of aerosol particle measurements at the high-altitude site Chacaltaya in Bolivia. Deployed instruments were one NAIS and an SMPS which allow the determination of time series of particles larger than 50, 80, and 100 nm. These fractions of the full particle size distribution are considered proxies for high, intermediate and low number concentrations of CCN. The data are then interpreted in terms of CCN formation from freshly nucleated particles (new particle formation NPF). Both the long duration of measurements and the unique measurement site make the data set highly valuable and justify publication in ACP. However, the authors have not fully explored the data set and use simplified assumptions in their analysis which requires major revisions. In addition, the structure of the paper can be significantly improved in order to help the reader to better getting the key messages of the study.

The key concerns can be summarized in three questions, which have mostly been addressed already by Anonymous Referee #1. In that respect I will focus on the major concerns and on suggested modifications of the way the material is presented.

SPECIFIC QUESTIONS

1. How do the authors separate CCN formation from freshly nucleated particles and CCN formation from the growth of pre-existing particles during nucleation events? Here, I refer to the very detailed comments of Anonymous Referee #1. There is not much to add.

2. How robust is the treatment of advection of different air masses by the selected ap-proach? Are there other observables (e.g., trace gases) available which allow a more robust treatment of particle transport than the simple method deployed in the study? The authors use the hypothesis that similar particle number concentrations are trans-ported to the site during days with and without particle formation events. They state in Section 3.1.2 that at hours outside of NPF events, particle number concentrations were on average similar for event and non-event days. However, what is the variability of the particle background and does it depend on the wind direction where the air masses came from, etc.? In particular the variability of the particle background needs to be presented more quantitatively since this parameter determines the level of uncertainty of the reported CCN increases by NPF.

Concerning the structure, the presentation of results in Section 3.1 is confusing. The authors start with a detailed description of CCN production and list all obtained num-bers in detail and show them in Fig. 2. Then in Section 3.1.2 they introduce a correction

of the presented CCN number concentrations. It is confusing that the CCN production neglecting the influence of advection is shown in Fig. 2 while the more important CCN production from NPF only is not shown but only listed in Table 2. If I understood right, the authors focus on CCN from NPF. If this is true then the way of presenting the data in Section 3.1 should be revised. One possibility is to start with a quantitative analysis of the "particle background" during non-event days, including its variability, introduce then the method for determining CCN production and present finally the CCN production values corrected for particle transport.

3. How robust is the separation between air masses form the boundary layer and from the free troposphere, and what is the expected impact of air mass history on the occurrence of new particle formation events? This question refers to Section 3.2 which in its current form is difficult to understand. The attempt of the authors is quite understandably to study if NPF events occur preferably in air masses originating from the BL or from the FT. However, doing this requires a clear presentation of event types and characteristics before going into details. Here the authors should restructure Section 3.2, start with a clear presentation of event types and scenarios. One table including all considered cases (with more detail than stated in Table 3) etc. might help. Looking at Fig. 5, there is no big difference between the scenarios, except for S1, S6 and likely S7. The authors may rethink the choice of scenarios in order to get a more precise conclusion from this part of the study. In addition, the expected impact of air mass history should be investigated / discussed.

MINOR COMMENTS

1. Since the classification of NPF events is crucial for understanding the manuscript, a brief description of types should be given at the end of section 2.2, instead of referring to the references Hirsikko et al. (2007) and Rose et al. (2015).

2. Please add brief descriptions of quantities J and GR to x axis of Figure 6.

---

## Author Comment (AC1) · 6 Dec 2016

We thank Referee N°1 for his very helpful comments and suggestions. We feel they have greatly improved our manuscript. We have addressed the comments below.

**Comment 1**: Introduction: The introduction is well written; however, I think that some major studies have been forgotten or omitted. Recently, studies conducted at the Jungfraujoch appeared in Science and JGR-A. These studies should be mentioned as a comparison, especially in terms of CCN production, would be extremely valuable (Herrmann et al. 2015, Bianchi et al. 2016, Tröstl et al. 2016)

**Reply 1**: Some of these studies are very recent and were not published before the initial submission of the present study to ACPD. They are now mentioned in the introduction, as well as in the discussion:

Introduction: "However, observations to validate these predictions are scarce, especially for the FT, where measurements are often technically challenging. Recent studies conducted at the Jungfraujoch station (Switzerland, 3580 m a.s.l.) reported significant enhancement of the particle concentration below 50 nm by NPF in the FT, while only a minor fraction of this particles grow beyond 90 nm, even on a time scale of several days (Herrmann et al., 2015; Tröstl et al., 2016). The contribution of NPF to the production of CCN is thus likely to be very limited in this part of the FT, while boundary layer originating particles were observed to dominate the CCN concentrations measured at Jungfraujoch. The occurrence of the NPF process itself in the FT was reported to be tightly connected with the strength of boundary layer influence at the site, together with global radiation (Bianchi et al., 2016; Tröstl et al., 2016)."

**Comment 2**: Page 2 Line 29: The reference Yli-Juuti is only about one site (Hyytiälä). I would recommend to consider the Manninen et al. (2010) EUCAARI paper which reports findings from 12 European sites.**

**Reply 2**: It is true that the study by Yli-Juuti et al. investigate measurements from Hyytiälä. However, it provides a very useful comparison between the GRs obtained at various locations, including all observations by Manninen et al. 2010 along with other additional studies.

**Comment 3*: Page 2 Line 30: Reference is needed.**

Reply 3: the synthesis by Kerminen et al. (2012) is now mentioned.

**Comment 4**: Page 3 Line 17: "....However, observations to validate these predictions are scarce, especially for the FT...." It is true that little information is available in this specific field of research. However, new study by Tröstl et al., (2016) has just been published in the Journal of Geophysical Research (Atmospheres) which (among other things) investigates the contribution of new particle formation to the CCN concentration in the Alps in some detail. I think a comparison to this work (Alps vs Andes) would be quite interesting, especially considering the general scarcity of similar research.

**Reply 4**: We do agree, however the paper by Tröstl et al. (2016) was not published (available online on September 8th) when the present study was submitted to ACPD (August 15th). The results from Jungfraujoch are now mentioned (See Reply 1).

**Comment 5**: Page 4 Line 1,2: The authors underline the vicinity of the site to a city like La Paz that has a large population and is assumingly rather polluted. This fact seems weirdly underused in the study. Why not determine La Paz air masses to find out what (if any) effect polluted air masses have on NPF and CCN production? A backtrajectory analysis might actually be quite interesting.

**Reply 5**: The effect of wind direction and air mass back trajectories on NPF occurrence and characteristics was already investigated by Rose et al. (2015b). NPF events are more frequent and also more intense during the dry season, in dominant north western winds corresponding to oceanic air masses (Fig. 11 of the mentioned study). The fastest particle growth, which are observed during the

west season, are in contrast observed in air masses that passed over the Amazon basin (Fig. 14 of the mentioned study). No distinctive feature connected to air masses from La Paz came out of this analysis. This is now shown on Fig S4 in the supplementary, and explicitly mentioned in Section 3.1.2:

"It is worth noticing that winds originating from the more polluted sector of La Paz – El Alto (south) do not seem to be over-represented neither on event nor on non-event days. However, because of the close proximity of this area, it is complex to further assess how it contributes to CCN concentration from wind direction alone, and we cannot exclude a bias related to the variability of this specific source between event and non-event days."

**Comment 6:** Section 2.2 Indirect method for the estimation of the NPF contribution to the CCN production I'm not sure that this method fulfilled what the authors claimed. I agree with the fact that the CCN increased during NPF is not just a pure coincidence since the NPF precursors certainly also facilitate growth. However, it's not possible to distinguish the CCN formed by the NPF events with the growth of pre-existing particles during the same time. An easy way to fix that somehow would be to assume that ALL pre-existing particles to become CCN before any new particles. I.e. the number of particles below 100 nm before the event must be subtracted from the CCN100 you are now using and so far for the other sizes. This still would not account for, say, 90 nm particles that are transported to the site during NPF and grow above the threshold and contribute to CCNmax but it would be better than the current approach.

**Comment 8**: Page 5 Line 1: "...The CCN production during an event was obtained from the comparison of the CCN concentration Ninit prior to and the maximum CCN concentration Nmax during the event..." I agree with the authors that this is the CCN production during an event. However, the authors treat this as the CCN increase due to NPF. As already mentioned, CCN increase DURING and BECAUSE OF NPF are not the same thing. This distinction demands for clarity of concept and language whenever the topic is discussed. The manuscript in its current form, however, conflates those things. Besides the need for exact language, I actually think the issue can be addressed (to an extent) as lined out above.

**Comment 9**: Page 5 Line4 "....tinit, when nucleated particles reach the threshold size...." I don't think is possible to know that nucleated particles reached CCN size instead of larger particles that simply have grown above the threshold. The respective figure1 actually shows that t\_init isn't found as the text claims: if the text was true, then t\_init\_100 should be well after t\_init\_50 because growth takes time. In the figure, however, all t\_init are the same. That means t\_init is really just the time when CCN numbers start to increase. But that increase doesn't likely come from NPF. Figure 1 illustrates a further problem with this claim: t\_init\_100 is roughly 1.5...2 h after nucleation onset. If those were really newly nucleated particles, we would need growth rates of 50 nm/h. I find that hard to believe as such numbers have never (to my knowledge) been reported in the literature for atmospheric nucleation.

Comments 6, 8 and 9 were addressed together since they all refer to the same topic.

**Reply 6 – 8 - 9**: Our approach to deal with the growth to CCN size of particles that are not issued from NPF is to substract the CCN increase observed during non NPF days from the CCN increase observed during NPF days (section 3.1.2.). However, we first used in section 3.1.1 the classical approach found in most papers dealing with the evaluation of CCN production from NPF, in order to be able to compare with existing values reported in the literature from other sites. We do agree that using this first analysis could be confusing if not well explained. We now use a different terminology for section 3.1.1, using the term « CCN production DURING NPF » instead of CCN production FROM NPF. Also, in order to make our approach clearer the methodologies and discussion on the uncertainties on these methodologies are included in the results sections.

If we understand correctly the second part of Comment 6, the reviewer suggests, for each threshold size D, to subtract the concentration of particles below D measured before the event to what we actually calculate as CCND. We believe that this methodology does not provide significant progress compared to what we did for two main reasons:

- This would only remove the particles that arrived at the station before the events, assuming that their concentration remains the same during the event. In any case, the contribution of the particles which reach the station during the event and further grow would still not be filtered out.
- This suggestion relies on the strong assumption that all particles below threshold D, regardless their origin, will reach D, which is unrealistic due to coagulation process.

In brief, we believe that applying such a methodology would consist to assume different hypothesis, which might not be more suitable than ours to describe what is actually occurring.

**Comment 7**: Page 4 Line 28: I found CCNhigh and CCNlow quite confusing. I would rather prefer to use the size of the particles, therefore, I would call them CCN50 and CCN100.

**Reply 7**: Notations were changed accordingly.**

**Comment 10**: Page 5 Line 14: The authors acknowledge only partially the previous point. They say that the particles can be transported during that period but they still don't mention that small particles transported there can then grow to the threshold and being considered as formed by NPF. They also correct the transported particles by comparing NPF days with non-NPF days. This assumption is valid only if the physic dynamic is the same. If the NPF is triggered by the wind convection might be that during nucleation (more wind) the particles transported up there are more. This point needs further investigation or at least being commented.

**Reply 10**: The possibility for particles formed off-site and transported to the station to contribute to CCN concentrations is now explicitly mentioned (see Reply 6 - 8 - 9).

The revised version of the manuscript now includes additional discussion regarding environmental conditions on event and non-event days, and the effect in can have when estimating transport contribution to CCN concentrations (Section 3.1.2):

"These calculations rely on the hypothesis that the specific environmental conditions on which NPF occurs are not influencing the transport from lower atmospheric layers. In order to further evaluate the reliability of this assumption, wind direction and speed as well as global radiation were investigated on event and non-event days (Figures S3 and S4 in the supplementary material). As previously reported by Rose et al. (2015a), NPF events are favoured during clear sky conditions, when radiation is higher (Fig. S3). Thus, there is likely a bias towards an underestimation of radiative driven transport from lower atmospheric layers due to the fact that cloudy days are over-represented for non-event days. Regarding wind, contrasting directions are also observed between event and non-event days (Fig. S4), with patterns closely related to those observed for the dry and wet seasons, respectively (Rose et al., 2015a). It is worth noticing that winds originating from the more polluted sector of La Paz – El Alto (south) do not seem to be over-represented neither on event nor on non-event days. However, because of the close proximity of this area, it is complex to further assess how it contributes to CCN concentration from wind direction alone, and we cannot exclude a bias related to the variability of this specific source between event and non-event days. Nonetheless, the particle number concentrations observed at the time preceding the usual occurrence of the NPF events are similar for event and nonevent days (Fig 3, S1, S2). Moreover, higher wind speeds are on average recorded on non-event days, that likely lead to an enhanced transport of particles to the site compared to event days, and hence lead to an underestimation of the contribution of NPF to the increase of CCN. In any case, taking into account the contribution of transport when calculating the increase of CCN concentrations after NPF events was never done in the past, and certainly helps approaching a more realistic view of the real contribution of NPF to CCN number concentrations.

Two additional figures are provided in the Supplementary material to show wind speed and direction (Fig. S4) and radiation (Fig. S3).

**Comment 11**: Section 2.3 Method to assess the influence of the boundary layer in Chacaltaya. To my understanding, this method only takes into account the local PBL influence at the time of nucleation. It does little to actually describe the air mass in which nucleation occurs. Bianchi et al. (2016) have shown that strong PBL contact 1-2 days before NPF is crucial in the case of the Alps (Jungfraujoch). While conditions are certainly different in the Andes, there is no reason to believe that local wind conditions could accurately describe an air mass and its history which is what one must do to get a handle on PBL influence. There is a good body of literature dealing with the assessment of PBL influence. Much of it has been summarized in recent papers by Bukowiecki et al. (2016) and Herrmann et al. (2015).

**Reply 11**: Indeed the methodology that we propose only takes into account the local BL influence at the time of the event. But this is obviously an interesting parameter to take into account since we do observe different event types as a function of the different evolution patterns of the BL (Fig. 6). The fact that only local influence is considered is now clearly stated in the title of Section 2.3 "Method to assess the local influence of the boundary layer in Chacaltaya", as well as in the introducing lines of the same section "In order to assess whether the site is under the influence of the planetary boundary layer or the low free troposphere at a local scale, regardless the history of the air mass, we employed the hourly-averaged value of the standard deviation of the horizontal wind direction ( $\sigma_{\theta}$ )".

Also, the valuable additional information which could be provided by an analysis of the air mass history (not performed here due to lack of proxy measurement and modelling tools such as those used at Jungfraujoch) is discussed at the end of Section 3.2.3, in the light of the work by Tröstl et al. (2016) and Bianchi et al. (2016):

"Additional analysis regarding the history of the air mass and BL influence along its trajectory would provide valuable information to even more assess the role of the exchanges between the BL and the FT on the occurrence of NPF and its contribution to the formation of new CCN. Indeed, observations conducted at the Jungfraujoch showed that stronger NPF events (type I) occurred in air masses one or two days after contact with the BL (Bianchi et al., 2016; Tröstl et al., 2016). These results are however based on proxies (CO, NOy) and modelling tools which were unfortunately not available for Chacaltaya. Nevertheless, our results goes to some extent into the same direction as the work by Tröstl et al. (2016) and Bianchi et al. (2016), at least supporting the major role of BL intrusion (regardless of its kind, before or during the event) to sustain particle growth. Similar FT feeding process from the BL was also shown by Rose et al. (2015b) at the puy de Dôme (France, 1465 m a.s.l.)."

**Comment 12**: Page 8 Line 14: "...when particles reached the lowest activation diameter, i.e. 50 nm, they systematically grew up to at least 100 nm..." This statement is stronger than what the data seem to support. We don't know for certain that those are not pre-existing particles that simply did a bit of growing above the considered threshold, or do we?

**Reply 12**: As previously mentioned in reply 6 - 8 - 9, the contribution of pre-existing particles cannot be excluded. Rephrasing and additional explanation should however help in this particular case:

"Over the whole year, 61% of the studied NPF events were apparently growing to CCN-relevant sizes, and when observed, the contribution of growing particles to CCN concentrations was systematically seen up to at least 100 nm. During the wet season, the frequency of aerosol particles reaching CCN sizes during a NPF event was higher compared to the dry season (79% and 56%, respectively). This last observation can be ascribed to the larger growth rates which were detected during the wet season, being on average enhanced by a factor 1.7 compared to the dry season (Rose et al., 2015a). It is however worth noticing that at this stage, the contribution of pre-existing particles transported to the site at already grown sizes cannot be excluded."

**Comment 13**: Page 8 Line 16: "...aerosol particles originating from NPF event and reaching CCN sizes..." Yet again the same problem that I think it should be fixed. I haven't seen any evidence that all those new CCN come directly from NPF, and, indeed, I find it highly unlikely: as long as there are pre-existing particles their chances to add to the CCN concentration are MUCH higher than the chances of newly formed particles.

**Reply 13**: Yes, we rephrased: "During the wet season, the frequency of aerosol particles reaching CCN sizes during a NPF event was higher compared to the dry season".

**Comment 14**: Page 9 from Line 15 to Line 32: The paragraph lacking a message. First the authors give us comparisons to sites that are hardly comparable to a 5000 m peak, and then they tell us in the last few lines that those comparisons are more or less pointless and I actually i would agree because these sites are just different and the comparison does not provide useful information. A comparison to Tröstl et al., (2016) might be more interesting, especially since those results are quite different.

**Reply 14**: The studies that are mentioned in the initial version of the manuscript were, at that time, the only ones that focussed on CCN production from NPF using a method that is similar to ours. A comparison with the recent results by Tröstl et al. (2016) is now included:

"The potential of NPF to contribute to CCN production at high altitude was more particularly investigated by Tröstl et al. (2016) at the Jungfraujoch station. Tröstl and co-workers found that newly formed particles did not directly grow to CCN sizes (90 nm at Jungfraujoch) within observable time scale (up to two days) but rather experienced a multi-step growth process over several days. As a consequence, the contribution of NPF to the CCN budget was complex to distinguish from that of other sources such as BL entrainment of larger particles, which was likely the main source of measured CCN."

**Comment 15**: Page 10 Line 8: Sunrise is typically a well-defined point in time and not a process that has an onset.

Reply 15: Changed to "that takes place after sunrise".

**Comment 16**: Section 3.1.2 Correction for the contribution of particles transported to the site I'm a bit concerned regarding this method to correct the contribution of particles transported. As mentioned earlier, this method is valid only in case every day we have the same physics and nucleation only depends on the vapors present. However, if nucleation is triggered by the wind coming up the valley than during nucleation we would have more transportation of big particles and therefore the correction method is not ideal. Would be nice to know what are the differences (Wind direction, wind speed etc..) during nucleation and during no nucleation where these background values is taken in account.

Reply 16: This point is now discussed in Section 3.1.2, in the light of supplementary Figure S4.:

"Regarding wind, contrasting directions are also observed between event and non-event days (Fig. S4), with patterns closely related to those observed for the dry and wet seasons, respectively (Rose et al.,

2015a). It is worth noticing that winds originating from the more polluted sector of La Paz – El Alto (south) do not seem to be over-represented neither on event nor on non-event days. However, because of the close proximity of this area, it is complex to further assess how it contributes to CCN concentration from wind direction alone, and we cannot exclude a bias related to the variability of this specific source between event and non-event days. Nonetheless, the particle number concentrations observed at the time preceding the usual occurrence of the NPF events are similar for event and non-event days (Fig 3, S1, S2). Moreover, higher wind speeds are on average recorded on non-event days, that likely lead to an enhanced transport of particles to the site compared to event days, and hence lead to an underestimation of the contribution of NPF to the increase of CCN."

**Comment 17**: Section 3.2 How layering influences growth to CCN-sizes I do understand the need of knowing where the nucleation events take place and especially if this lead to a big production of particles in the free troposphere. However, I believe that dividing in 10 scenarios is a bit over exaggerated and probably not quite realistic. I think it would be better if the authors can simplify this section. I don't think that selecting more than 3 scenarios is feasible. In addition to that the split into different scenarios seems illconceived since most scenarios are quite irrelevant with very little occurrences. This might all be a nice exercise in data analysis but the text fails to tell us what the actual results are. What do we learn in this section apart from some minuscule details? This section has the feel of filler material and needs to be improved with a fair amount of actual substance.

**Reply 17**: Sections 3.2.1 and 3.2.2 were simplified, with a reduced number of scenarios (3) focussed on BL and FT tropospheric conditions, which are the most frequent.

**Comment 18**: Page 12 Line 4: "...regarding the location of the station in the tropospheric layers..." I wonder if this is actually relevant at all? The NPF events are mainly driven by the air mass history and not so much by the atmospheric layer when the event begins.**

**Reply 18**: If air mass history has an influence on the nucleation process, based on Fig. 6 it seems that local atmospheric layering at the station during NPF does also have as effect, since different event types are observed as a function of the different evolution pattern of the BL. For example, we clearly observe that for events triggered in the FT, the probability for type I events to occur is increased when initial free tropospheric conditions are changing in the course of the events. Additional discussion on air mass history is now provided at the end of Section 3.2.3.

**Comment 19: Page 12 Line 8: "389 NPF events" Is that a different data set?**

Reply 19: The events included in the CCN production investigation are only those belonging to class I, as stated P8, L10: "147 days showing type I NPF events". In contrast, the analysis regarding the influence of atmospheric layering on the occurrence of NPF (Sections 3.2.1 and 3.2.2) includes all the events previously discussed by (Rose et al., 2015b), not only type I events. It is now clearly stated: "389 NPF events (including all event types, i.e. I, II or bump, Hirsikko et al., 2007) previously discussed by (Rose et al., 2015a)..."

Minor edits:

**Figures: In general no need to state Chacaltaya at the end of every captions**

Ok, removed.

Figure 1: Why Tinit is not before the nucleation but already a after the start of the event? Please also describe the figure, Particle size distribution measured by..... and so on.

P5, L9:  $t_{init}$  is defined as the time " when nucleated particles reach the threshold size", or let's say growing particles, since their origin might be uncertain. In other words,  $t_{init}$  is the time when the "banana" reach the threshold size, further leading to the CCN concentration increase, as shown on Fig. 1.

Additional information is provided: "Determination of the CCN concentration increase for the 3 threshold diameters (50, 80 and 100 nm) from the particle size distribution measured by SMPS".

---

## Author Comment (AC2) · 6 Dec 2016

We thank referee N°2 for his comments and remarks which contributed to improve and clarify the present paper. Our answers to the suggestions are listed below.

**SPECIFIC COMMENTS**

*Comment 1: How do the authors separate CCN formation from freshly nucleated particles and CCN formation from the growth of pre-existing particles during nucleation events? Here, I refer to the very detailed comments of Anonymous Referee #1. There is not much to add.*

**Reply 1**: We agree with the fact that the manuscript needed for more clarity regarding this aspect. We now use a different terminology for section 3.1.1, using the term « CCN production DURING NPF » instead of CCN production FROM NPF. Also, in order to make our approach clearer the methodologies and discussion on the uncertainties on these methodologies are included in the results sections. In addition, several sentences were rephrased throughout the manuscript in order to further avoid any misunderstanding.

*Comment 2: How robust is the treatment of advection of different air masses by the selected approach? Are there other observables (e.g., trace gases) available which allow a more robust treatment of particle transport than the simple method deployed in the study? The authors use the hypothesis that similar particle number concentrations are transported to the site during days with and without particle formation events. They state in Section 3.1.2 that at hours outside of NPF events, particle number concentrations were on average similar for event and non-event days. However, what is the variability of the particle background and does it depend on the wind direction where the air masses came from, etc.? In particular the variability of the particle background needs to be presented more quantitatively since this parameter determines the level of uncertainty of the reported CCN increases by NPF. Concerning the structure, the presentation of results in Section 3.1 is confusing. The authors start with a detailed description of CCN production and list all obtained numbers in detail and show the min Fig. 2. Then in Section 3.1.2 they introduce a correction of the presented CCN number concentrations. It is confusing that the CCN production neglecting the influence of advection is shown in Fig. 2 while the more important CCN production from NPF only is not shown but only listed in Table 2. If I understood right, the authors focus on CCN from NPF. If this is true then the way of presenting the data in Section 3.1 should be revised. One possibility is to start with a quantitative analysis of the "particle background" during non-event days, including its variability, introduce then the method for determining CCN production and present finally the CCN production values corrected for particle transport.*

**Reply 2**: We know that the correction that we applied for particle transport is not completely accurate, since it relies on a strong hypothesis which is explicitly mentioned in the text ("similar number concentrations of particles are transported to the site on event and non-event days"), and which might actually not be verified because of some reasons which are now mentioned with more details in the revised version of the manuscript (Section 3.1.2):

"These calculations rely on the hypothesis that the specific environmental conditions on which NPF occurs are not influencing the transport from lower atmospheric layers. In order to further evaluate the reliability of this assumption, wind direction and speed as well as global radiation were investigated on event and non-event days (Figures S3 and S4 in the supplementary material). As previously reported by Rose et al. (2015a), NPF events are favoured during clear sky conditions, when radiation is higher (Fig. S3). Thus, there is likely a bias towards an underestimation of radiative driven transport from lower atmospheric layers due to the fact that cloudy days are over-represented for non-event days. Regarding wind, contrasting directions are also observed between event and non-event days (Fig. S4), with patterns closely related to those observed for the dry and wet seasons, respectively (Rose et al., 2015a). It is worth noticing that winds originating from the more polluted sector of La Paz – El Alto

(south) do not seem to be over-represented neither on event nor on non-event days. However, because of the close proximity of this area, it is complex to further assess how it contributes to CCN concentration from wind direction alone, and we cannot exclude a bias related to the variability of this specific source between event and non-event days. Nonetheless, the particle number concentrations observed at the time preceding the usual occurrence of the NPF events are similar for event and non-event days (Fig 3, S1, S2). Moreover, higher wind speeds are on average recorded on non-event days, that likely lead to an enhanced transport of particles to the site compared to event days, and hence lead to an underestimation of the contribution of NPF to the increase of CCN. In any case, taking into account the contribution of transport when calculating the increase of CCN concentrations after NPF events was never done in the past, and certainly helps approaching a more realistic view of the real contribution of NPF to CCN number concentrations."

Additional figures are also provided in the supplementary, showing wind speed and direction (Fig. S4) and radiation (Fig. S3)

However, the aim of this correction is not to provide an exact estimation of transport contribution, which, we believe, cannot be retrieved from these measurements, even if including additional parameters, such as trace gases concentrations. Our objective, less ambitious, is rather to go one step further compared with previous analysis published in the literature in order to estimate NPF contributions to CCN number concentrations which are closer to actual values, as mentioned in the text.

We agree that the terminology to describe sections 3.1.1 and 3.1.2. was confusing. We now use the term « CCN formation during NPF » for section 3.1.1 and CCN formation from NPF for section 3.1.2.

We agree that the most important results should be better emphasised for section 3.1.2. For that purpose, an additional Figure (4) is now provided and the text has been slightly changed:

"The contributions of NPF particles to the increase of CCN, all shown on Fig. 4.a. and reported in Table 2 for the different seasons and sizes, hence represent a significant fraction of the CCN increase shown on Fig. 2.a. and reported in Table 1. The contribution of NPF to CCN concentrations are comparable or even higher than those previously mentioned for other stations in the literature, which probably also include CCN sources other than NPF. The relative impact of NPF are estimated to increase the $CCN_{50}$ number concentrations by more than 250 % during both seasons, and the $CCN_{100}$ number concentrations by more than 100% and 200% during the dry season and wet season, respectively."

*Comment 3: How robust is the separation between air masses form the boundary layer and from the free troposphere, and what is the expected impact of air mass history on the occurrence of new particle formation events? This question refers to Section 3.2 which in its current form is difficult to understand. The attempt of the authors is quite understandably to study if NPF events occur preferably in air masses originating from the BL or from the FT. However, doing this requires a clear presentation of event types and characteristics before going into details. Here the authors should restructure Section 3.2, start with a clear presentation of event types and scenarios. One table including all considered cases (with more detail than stated in Table 3) etc. might help. Looking at Fig. 5, there is no big difference between the scenarios, except for S1, S6 and likely S7. The authors may rethink the choice of scenarios in order to get a more precise conclusion from this part of the study. In addition, the expected impact of air mass history should be investigated / discussed.*

**Reply 3**:

The method used to distinguish between boundary layer and free troposphere air masses performed in the present study is the only one we could apply given the measurements conducted in Chacaltaya.

The impact of air mass history, including occurrence and length of BL contact, has not been investigated due to lack of proxy measurement and modelling tools (such as those used by Tröstl et al. (2016)). However, the fact that air mass history may, in addition to local conditions, influence the occurrence of NPF is now mentioned at the end of Section 3.2.3 in the light of the work by Tröstl et al. (2016) and Bianchi et al. (2016):

"Additional analysis regarding the history of the air mass and BL influence along its trajectory would provide valuable information to even more assess the role of the exchanges between the BL and the FT on the occurrence of NPF and its contribution to the formation of new CCN. Indeed, observations conducted at the Jungfraujoch showed that stronger NPF events (type I) occurred in air masses one or two days after contact with the BL (Bianchi et al., 2016; Tröstl et al., 2016). These results are however based on proxies (CO, NOy) and modelling tools which were unfortunately not available for Chacaltaya. Nevertheless, our results goes to some extent into the same direction as the work by Tröstl et al. (2016) and Bianchi et al. (2016), at least supporting the major role of BL intrusion (regardless of its kind, before or during the event) to sustain particle growth. Similar FT feeding process from the BL was also shown by Rose et al. (2015b) at the puy de Dôme (France, 1465 m a.s.l.)."

We agree with the fact that the number of scenarios used to assess the impact of BL on the occurrence of NPF was maybe too high. It was thus reduced in order to clarify the message:

" The most frequent scenarios, which include more than 88% of the documented events, are listed, together with their frequency of occurrence, in Table 3. Scenario S1 refers to those days when the first steps of the NPF process were observed to occur in the BL, while scenario S2 refer to the events started in the FT. Scenario S2 is further divided into two sub-classes to distinguish between the events which first steps occur exclusively in the FT (S2.1) from those during which BL dynamics lead to changing conditions in the course of the event (S2.2). Events triggered in the IL are not frequently observed compared to those initiated in the BL or in the FT, and are thus not highlighted in this classification." (Section 3.2.1)

The description of the scenarios in Table 3 was also detailed.

**MINOR COMMENTS**

*Comment 1: Since the classification of NPF events is crucial for understanding the manuscript, a brief description of types should be given at the end of section 2.2, instead of referring to the references Hirsikko et al. (2007) and Rose et al. (2015).*

**Reply 1**: As suggested, brief description of the event types is now provided in Section 3.1.1:

"First, only those NPF events referred as type I, i.e. with clear particle growth from smallest sizes, were considered; they contrast with type II events, during which the growth is more irregular and may be interrupted in certain size ranges, and bump type events, which completely miss the growth of the newly formed clusters (Hirsikko et al., 2007; Yli-Juuti et al., 2009)."

*Comment 2: Please add brief descriptions of quantities J and GR to x axis of Figure 6.*

**Reply 2**: Figure caption was slightly modified:

"Median formation rate of 2 nm particles ($J_2$) and growth rate in the range 1-3 nm ($GR_{1-3}$) reported separately for type I events initiated in the BL (scenario S1) and in the FT (scenario S2). Lower and upper limits of the error bars stand for the 1st and 3rd quartile, respectively."

Additional information is also provided in the text, before the description of Fig. 7:

"In order to further characterize the NPF events in the different atmospheric layers, statistics regarding the formation rate of 2 nm particle and the growth rate (GR) in the size range 1-3 nm as a function of the scenarios were performed for type I events. Growth rates were derived from the particle number size distribution using the "maximum" method from Hirsikko et al. (2005), while formation rates were calculated according to Kulmala et al. (2007)."